# The effect of de-escalation of P2Y$_{12}$ receptor inhibitor therapy after acute myocardial infarction in patients undergoing percutaneous coronary intervention: A nationwide cohort study

Jong-Shiuan Yeh[1,2], Chien-Yi Hsu[2,3], Chun-Yao Huang[2,3], Wan-Ting Chen[4], Yi-Chen Hsieh[5,6,7], Li-Nien Chien[4,8]*

1 Division of Cardiovascular Medicine, Department of Internal Medicine, Taipei Municipal Wan-Fang Hospital, Taipei, Taiwan, 2 Department of Internal Medicine, School of Medicine, College of Medicine, Taipei Medical University, Taipei, Taiwan, 3 Division of Cardiology and Cardiovascular Research Center, Department of Internal Medicine, Taipei Medical University Hospital, Taipei, Taiwan, 4 Health and Clinical Research Data Center, Office of Data Science, Taipei Medical University, Taipei, Taiwan, 5 PHD Program of Neural Regenerative Medicine, College of Medical Science and Technology, Taipei Medical University, Taipei, Taiwan, 6 PHD Program in Biotechnology Research and Development, College of Pharmacy, Taipei Medical University, Taipei, Taiwan, 7 Master Program in Applied Molecular Epidemiology, College of Public Health, Taipei Medical University, Taipei, Taiwan, 8 School of Health Care Administration, College of Management, Taipei Medical University, Taipei, Taiwan

* lnchien@tmu.edu.tw

**Data Availability Statement:** Data are available from the Health and Welfare Science Data Center (HWDC), Ministry of Health and Welfare in Taiwan.

## Abstract

To examine the effect of de-escalation of P2Y$_{12}$ inhibitor in dual antiplatelet therapy (DAPT) on major adverse cardiovascular events (MACE) and bleeding complications after acute myocardial infarction (AMI) in Taiwanese patients undergoing percutaneous coronary intervention (PCI). Patients who had received PCI during hospitalization for AMI (between 2013 and 2016) and were initially treated with aspirin and ticagrelor and without adverse events after 3 months of treatment were retrospectively evaluated. In total, 1,901 and 8,199 patients were identified as "de-escalated DAPT" (switched to aspirin and clopidogrel) and "unchanged DAPT" (continued on aspirin and ticagrelor) cohorts, respectively. With a mean follow-up of 8 months, the incidence rates (per 100 person-year) of death, AMI readmission and MACE were 2.89, 3.68, and 4.91 in the de-escalated cohort and 2.42, 3.28, and 4.72 in the unchanged cohort, respectively, based on an inverse probability of treatment weighted approach that adjusting for baseline characteristics of the patients. Multivariate Cox regression analyses showed the two groups had no significant differences in the hazard risk of death, AMI admission, and MACE. Additionally, there was no observed difference in the risk of bleeding, including major or clinically relevant non-major bleeding. The real-world data revealed that de-escalation of P2Y$_{12}$ inhibitor in DAPT was not associated with a higher risk of death or AMI readmission in Taiwanese patients with AMI undergoing successful PCI.

Due to legal restrictions imposed by the government of Taiwan in relation to the Personal Information Protection Act, data cannot be made publicly available. Requests for data can be sent as a formal proposal to the HWDC with an IRB approval for research purpose only. To access the data, please contact the Health and Welfare Science Data Center (stsung@mohw.gov.tw; stpeicih@mohw.gov.tw).

**Funding:** This study was supported by Health and Clinical Research Data Center, Office of Data Science, Taipei Medical University. The funder had no role in study design, data collection and analysis, decision to publish, or preparation of the manuscript.

**Competing interests:** The authors have declared that no competing interests exist.

## Introduction

Dual antiplatelet therapy (DAPT) with aspirin and clopidogrel has been recommended for greater than 10 years as the gold standard for antithrombotic therapy for patients with acute coronary syndrome (ACS). Ticagrelor is a newer generation of oral P2Y$_{12}$-receptor inhibitors, approved in 2011 by the Food and Drug Administration [1]. In Taiwan, ticagrelor has been approved and reimbursed for ACS patients by the Taiwan National Health Insurance System since 2013. Ticagrelor has a potent, faster-acting, and more predictable antiplatelet effect compared with clopidogrel, which translates into improved clinical outcomes in patients with ACS, despite an increased risk of bleeding. Modification of the oral P2Y$_{12}$ inhibitor regimen in order to prevent ischemic events with acceptable bleeding risk in patients with ACS is challenging and requires intensive research [2].

Switching between oral P2Y$_{12}$ inhibitors can enhance or reduce the degree of P2Y$_{12}$ receptor inhibition [3]; known as DAPT escalation and de-escalation, respectively. In the pivotal PLATO trial, ticagrelor significantly reduced ischemic events, especially in the early period after percutaneous coronary intervention (PCI) when compared with clopidogrel [4]. However, bleeding complications were inevitable during the maintenance phase of DAPT. The P2Y$_{12}$ inhibitor de-escalation strategy is already considered and used by many physicians when treating patients with ACS in order to reduce further bleeding risks [5]. Recently, both unguided (platelet function testing independent) and guided (platelet function testing dependent or CYP2C19 genotype guided) P2Y$_{12}$ inhibitor de-escalation strategies have been investigated in several clinical studies, however the data remain limited and conflicting [6–8]. Thus, the objective of this study was to examine the effect of de-escalated P2Y$_{12}$ inhibitor switching in DAPT on the major cardiovascular risks in patients with acute myocardial infarction (AMI) undergoing PCI based on real-world data from the Taiwan's National Health Insurance Research Database (NHIRD).

## Methods

### Ethics statement and data source

In this retrospective and population-based cohort study, we used the NHIRD that provided by Health and Welfare Science Data Center (HWDC), Ministry of Health and Welfare in Taiwan. The HWDC is a third-party organization under a government initiative that allows Taiwanese researchers applying the access right to analyze 30+ health related databases. According to the regulation of HWDC, individual identifiers are encrypted to protect the privacy of beneficiaries and are released to investigators for research purposes. Therefore, all data were fully anonymized before we access them. The data can be used only in an independent HWDC operation zone and only statistical results can be brought out from the zone. Additionally, this study was approved by the Institutional Review Board of Taipei Medical University (TMU-JIRB N201903043) and ethics committee waived the requirement for informed consent.

### Study design

The NHIRD was established by the National Health Insurance Administration (NHIA) of Taiwan, which covers 99% of Taiwanese residents. Notably, the NHIRD contains diagnostic codes for the International Classification of Diseases Ninth Revision, Clinical Modification (ICD-9-CM), and the Tenth Revision (ICD-10) after 2016, treatment procedures, claims for prescribed drugs, service dates, reimbursement amounts, demographic information, as well as encrypted beneficiary and provider identifiers. Death records were obtained from the National

Death Registry. Two data sets were able to be linked together by using unique encrypted identifiers.

## Study population

The cohort was comprised with patients first hospitalized with a primary diagnosis of AMI (ICD-9-CM: 410) between July 1$^{st}$, 2013 and December 31$^{th}$, 2016. This period was chosen because ticagrelor was first approved by the NHIA for the treatment of AMI patients after July 1$^{st}$, 2013. The date of hospital admission following AMI was defined as the index date of AMI. Several exclusion criteria were applied: 1) patients aged less than 18 years, without information identifying sex, or not a citizen of Taiwan; 2) patients did not have heparin or antiplatelet agents or had aspirin only or had antiplatelet agents other than ticagrelor at index of AMI; 3) patients had undergone a coronary artery bypass graft during the study period. Because the DAPT prescription with either ticagrelor or clopidogrel in these patients were low despite the guidelines recommended [9], we therefore excluded these patients to increase the homogeneity of the study sample; 4) patients died within 3 months after the index of AMI. The later exclusion was made because their disease condition and comorbidities were more likely to be complicated; thus, their treatment were less likely to follow the recommended guidelines. Besides, we used a 3-month window to group patients into de-escalation cohort, resulting in being unable to classify these patients. Thus, we excluded this population. The patients treated with prasugrel were not included because it had not been approved for reimbursement under the regulation of Taiwanese National Health Insurance during the study period. Of the eligible patients, these who had no prescription for clopidogrel or ticagrelor within 3-month of follow-up were also excluded. In the analytic cohort, a 3-month window was used to identify whether their DAPT treatment scheme was de-escalated from ticagrelor plus aspirin to clopidogrel plus aspirin. A 3-month window was chosen because we can observe at least 2 outpatient visits in the period, allowing de-escalating of P2Y12 inhibitors if necessary. The patients were considered as the de-escalation group if ticagrelor was changed to clopidogrel within 3 months after index AMI hospitalization. Fig 1 presents the patient selection process.

## Inverse probability of treatment weighting

To account for potential selection bias, an inverse probability of treatment weighted (IPTW) approach to balance baseline differences between the two groups was used. This method has

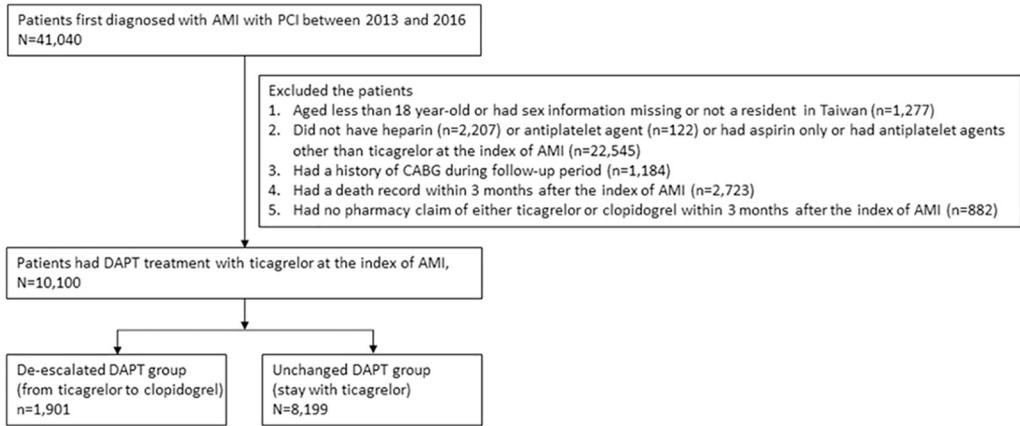

**Fig 1. Patient selection process.** Abbreviation: AMI = acute myocardial infarction; CABG = coronary artery bypass surgery; PCI = percutaneous coronary intervention.

been recommended in observational studies that compared different treatment alternatives, allowing an estimate for the relative treatment effect on time-to-event outcomes with minimal bias. The IPTW approach utilized the entire cohort and had the advantage of addressing a very large number of confounding variables instead of matching two treatment individuals on a select group of confounders. Covariates of gender, age, social economic status (SES), Charl-son-Deyo index, ORBIT score, medication passion ratio (MPR), procedure during AMI admission, comorbidities, and medication before the index of AMI in order to estimate weight were used (Table 1). Individuals were assigned a weight based on the likelihood of exposure to the treatment effect under investigation. The monthly income, occupation, and insurance listed in NHIRD were used to create a proxy for SES level. The Charlso-Deyo index was applied to adjust the severity of comorbidities between the two study groups. ORBIT score was evaluated and adjusted to mitigate the difference in bleeding risk between the two groups. We also considered the level of MPR, a measurement for drug adherence that may be associated with the efficacy of medication. Surgical procedure, including IABP, type of stent and complexity of PCI during AMI admission were also adjusted for confounding factors caused by different surgical procedures. We also considered comorbidities defined as more than 2 diagnostic claims made 1 year prior to the index date of AMI. Finally, common cardiovascular/bleeding medications were taken into account, including HMG-CoA reductase inhibitors, angiotensin-converting enzyme inhibitors/angiotensin II receptor blockers (ACEI/ARB), beta-blocker, anticoagulants, and aspirin.

## Main outcome measures

Three primary outcomes, including death, AMI readmission, and MACE, were assessed within one year during the follow-up period. AMI readmission was defined as the patient receiving a primary or secondary diagnosis of AMI after the index of AMI admission. MACE was considered to be CV death, non-fatal AMI hospital admission, and non-fatal stroke hospital admission. We also considered the risk of major bleeding, defined as gastrointestinal bleeding or other non-critical site bleeding that required transfusion of >2 units of packed red blood cells, or intracranial bleeding and the other critical site bleeding that led to hospitalizations. The risk of non-major clinically relevant bleeding was determined if the patient had an inpatient or outpatient visit for gastrointestinal and other non-critical site bleeding. This study assessed the effect of DAPT de-escalation strategy on the subsequent endpoints, therefore the patients with any event occurred before the 3 months DAPT de-escalation time point was censored. The accuracy of diagnostic code of AMI admission has been validated with a high PPV of 0.88 in any diagnosis based on NHIRD [10]. In this current study, we additionally combined antiplatelet therapy to confirm the AMI diagnosis was active. The accuracy of bleeding diagnosis has not been validated in NHIRD; however, a validation study conducted in Korea found that the PPV of primary diagnostic code of GI bleeding in Health Insurance Claim was over 90%, showing favorable reliability [11]. The data of CV death was derived from National Death Registry (NDR). The completeness and accuracy of death records in Taiwan were high [12], as it is mandatory to register all causes of death in the NDR.

The disease diagnosis codes for study outcomes are provided in S1 Table.

## Statistical analysis

Baseline characteristics were analysed using standardized mean differences (SMD). SMD is the most commonly used statistic to examine the balance of covariate distribution between treatment groups in the propensity score analysis. Because SMD is independent of the unit of measurement, it allows comparison between variables with different unit of measurement. An

**Table 1. Baseline characteristics of patients in unchanged DAPT group versus de-escalation DAPT group.**

| | Before IPTW | | | After IPTW | | |
|---|---|---|---|---|---|---|
| | Unchanged DAPT (n = 8,199) | De-escalated DAPT (n = 1,901) | SMD | Unchanged DAPT (n = 8,199) | De-escalated DAPT (n = 1,901) | SMD |
| Male | 85.9 | 79.4 | 0.17 | 84.6 | 84.5 | <0.01 |
| Age(years), mean ± SD | 59.0 ±12.5 | 62.5 ±13.1 | 0.27 | 59.6 ±12.8 | 60.0 ±12.5 | 0.03 |
| 18–64 | 68.8 | 56.6 | 0.26 | 66.5 | 66.5 | <0.01 |
| 65–74 | 18.4 | 22.1 | 0.09 | 19.1 | 18.9 | 0.01 |
| 75+ | 12.8 | 21.3 | 0.23 | 14.4 | 14.7 | 0.01 |
| SES level | | | | | | |
| 1 (Highest) | 8.2 | 8.4 | 0.01 | 8.3 | 8.2 | <0.01 |
| 2 | 8.6 | 7.8 | 0.03 | 8.6 | 7.6 | 0.04 |
| 3 | 44 | 45.6 | 0.03 | 43.9 | 45.8 | 0.04 |
| 4 | 37.4 | 36.8 | 0.01 | 37.4 | 36.8 | 0.01 |
| 5 (Lowest) | 1.8 | 1.4 | 0.03 | 1.8 | 1.5 | 0.02 |
| Charlson-Deyo index | 2.2 ±1.5 | 2.5 ±1.8 | 0.21 | 2.3 ±1.6 | 2.3 ±1.7 | 0.04 |
| 0–1 | 43.7 | 36.6 | 0.15 | 42.3 | 42.5 | <0.01 |
| 2+ | 56.3 | 63.4 | 0.15 | 57.7 | 57.5 | <0.01 |
| ORBIT score | 0.6 ±0.9 | 0.9 ±1.2 | 0.21 | 0.7 ±1.0 | 0.7 ±1.0 | 0.03 |
| 0–1 | 95.5 | 90.6 | 0.19 | 94.6 | 94.5 | <0.01 |
| 2+ | 4.5 | 9.4 | 0.19 | 5.4 | 5.5 | <0.01 |
| MPR | | | | | | |
| > = 0.80 | 75.7 | 71.2 | 0.10 | 74.8 | 74.5 | 0.01 |
| 0.4–0.8 | 9.8 | 10.4 | 0.02 | 9.9 | 10 | <0.01 |
| <0.40 | 14.5 | 18.4 | 0.10 | 15.3 | 15.5 | 0.01 |
| Procedure at index AMI | | | | | | |
| IABP | 4.9 | 3.9 | 0.05 | 4.7 | 4.9 | 0.01 |
| Complex PCI (> 1 vessels) | 5.2 | 4.3 | 0.04 | 5 | 5 | <0.01 |
| No. of stents, mean ± SD | 1.2 ±0.7 | 1.1 ±0.7 | 0.05 | 1.2 ±0.7 | 1.1 ±0.7 | <0.01 |
| Comorbidities, yes | | | | | | |
| Hypertension | 62.1 | 66.0 | 0.08 | 62.8 | 62.8 | <0.01 |
| Diabetes | 34.9 | 35.6 | 0.01 | 35.1 | 35.1 | <0.01 |
| Hyperlipidemia | 64.6 | 61.4 | 0.07 | 64 | 63.9 | <0.01 |
| Congestive heart failure | 15.8 | 21.1 | 0.14 | 16.8 | 16.8 | <0.01 |
| Ischemic stroke | 3.9 | 6.7 | 0.12 | 4.4 | 4.5 | <0.01 |
| Valvular heart disease | 3.7 | 5.7 | 0.10 | 4.1 | 4.1 | <0.01 |
| CLD | 3.5 | 4.8 | 0.07 | 3.8 | 3.7 | <0.01 |
| COPD | 5.6 | 9.0 | 0.13 | 6.3 | 6.3 | <0.01 |
| CKD | 15.4 | 18.6 | 0.09 | 16 | 16 | <0.01 |
| ICH | 0.3 | 0.6 | 0.04 | 0.4 | 0.3 | 0.01 |
| GI bleeding | 2.8 | 5.9 | 0.16 | 3.4 | 3.4 | <0.01 |
| Malignancy | 3.7 | 4.3 | 0.03 | 3.8 | 3.5 | 0.01 |
| Medication use, yes | | | | | | |
| Statin | 20.7 | 24.7 | 0.10 | 21.4 | 21.3 | <0.01 |
| ACEI | 5.6 | 6.2 | 0.02 | 5.7 | 5.6 | 0.01 |
| ARB | 26.1 | 31.8 | 0.13 | 27.2 | 27.3 | <0.01 |
| β-blocker | 20.5 | 27.8 | 0.17 | 21.9 | 22 | <0.01 |
| Anticoagulants | 0.8 | 1.2 | 0.04 | 0.9 | 0.8 | <0.01 |
| Aspirin | 18.4 | 24.10 | 0.14 | 19.50 | 19.40 | <0.01 |

*(Continued)*

**Table 1.** (Continued)

| | Before IPTW | | | After IPTW | | |
| --- | --- | --- | --- | --- | --- | --- |
| | Unchanged DAPT (n = 8,199) | De-escalated DAPT (n = 1,901) | SMD | Unchanged DAPT (n = 8,199) | De-escalated DAPT (n = 1,901) | SMD |
| PPIs | 2.9 | 4.6 | 0.09 | 3.3 | 3.4 | 0.01 |
| Follow-up period (Month), Mean [SD] | 7.2 ±3.1 | 8.1 ±3.6 | 0.28 | 7.2 ±3.1 | 8.2 ±3.5 | 0.29 |

**Abbreviations**: Values are % or mean SD ACEI = angiotension-converting enzyme inhibitors; ARB = angiotensin Receptor Blocker; Charlson-Deyo index = myocardial infarction, congestive heart failure, peripheral vascular disease, cerebrovascular disease, dementia, chronic pulmonary disease, rheumatologic disease, peptic ulcer disease, mild liver disease, diabetes, diabetes with chronic complications, hemiplegia or paraplegia, renal disease, moderate or severe liver disease, acquired immune deficiency syndrome; CLD = chronic liver disease; CKD = chronic kidney disease; COPD = chronic obstructive pulmonary disease; IABP = intra-aortic balloon pump; ICH = intracerebral hemorrhage; IPTW = inverse probability of treatment weighting; MPR = medication possession ratio; ORBIT = ORBIT score = age > = 74 years, anemia, bleeding history, chronic kidney disease, treatment with antiplatelet; SMD = standardized mean difference.

SMD of >0.1 indicated the presence of a non-negligible difference between the two groups [13]. The primary analysis was performed to evaluate the risk of major cardiovascular events and bleeding complications between the switched and unswitched group after IPTW. The cumulative incidence rate of each main outcome between groups was analyzed using Kaplan-Meier estimates. Cox proportional hazard regression was used to compare the risk of main outcomes between groups after adjustment for the variables shown in Table 1. The assumption of proportional hazard which assumes constant hazard ratio at any time point was evaluated and the analyses were not in violation. All analyses were performed using SAS/STAT 9.4 (SAS Institute Inc., Cary, NC, USA) and R package 3.5 (R core team, Vienna, Austria). Statistical significance was set at p< 0.05.

## Results

### Baseline characteristics

Among eligible patients, 1,901 were identified to be in the de-escalated DAPT group and 8,199 in the unchanged group. Baseline characteristics were analyzed using standardized mean differences (SMD). SMD is the most commonly used statistic to examine the balance of covariate distribution between treatment groups in the propensity score analysis. Because SMD is independent of the unit of measurement, it allows comparison between variables with different unit of measurement. An SMD of >0.1 indicated the presence of a non-negligible difference between the two groups [13]. Patients in the de-escalated group were mostly male, older, had higher Charlson-Deyo Score, ORBIT score and a history of hypertension, congestive heart failure, ischemic stroke, COPD and GI bleeding compared with those in the unchanged group. Patients in the de-escalated group were more likely to be prescribed ARB, beta-blockers, aspirin, and PPI. The 2 experimental groups had no difference in baseline characteristics (Table 1). The distribution of propensity score between 2 groups was provided in the supplementary materials.

### Survival analysis of main outcomes

The cumulative incidence rates of all cause death, AMI readmission, and MACE for the two groups (within a year of follow up) are shown in Table 2, Fig 2A–2C.

The incidence rates (per 100 person-years) of death, AMI readmission, and MACE were not significantly higher in the de-escalated group (2.89, 95% confidence interval [CI] = 2.05–

**Table 2. The incidence (per 100 PY) and adjusted HR of major vascular and bleeding events in unchanged and de-escalated DAPT group during one-year follow-up.**

| Outcomes | Group | No. of Event | PY | Incidence | (95% CI) | Adjusted* HR | (95% CI) | P-value |
|---|---|---|---|---|---|---|---|---|
| All cause death | Unchanged | 117 | 4,842 | 2.42 | (2.02–2.90) | 1.00 | (Ref.) | |
| | De-escalated | 37 | 1,274 | 2.89 | (2.05–3.91) | 1.20 | (0.83–1.73) | 0.336 |
| AMI hospitalization | Unchanged | 157 | 4,788 | 3.28 | (2.81–3.83) | 1.00 | (Ref.) | |
| | De-escalated | 46 | 1,256 | 3.68 | (2.75–4.88) | 1.12 | (0.80–1.56) | 0.509 |
| MACE | Unchanged | 226 | 4,783 | 4.72 | (4.13–5.36) | 1.00 | (Ref.) | |
| | De-escalated | 61 | 1,252 | 4.91 | (3.80–6.26) | 1.04 | (0.78–1.39) | 0.766 |
| Major bleeding | Unchanged | 114 | 4,821 | 2.36 | (1.95–2.82) | 1.00 | (Ref.) | |
| | De-escalated | 27 | 1,264 | 2.12 | (1.41–3.01) | 0.92 | (0.61–1.37) | 0.669 |
| Non-major clinically relevant bleeding | Unchanged | 688 | 4,605 | 14.95 | (13.8–16.1) | 1.00 | (Ref.) | |
| | De-escalated | 171 | 1,192 | 14.32 | (12.3–16.6) | 0.99 | (0.84–1.17) | 0.902 |

* Adjusted HR was estimated by Cox proportional regression controlling for covariates listed in Table 1.

Abbreviation: CI = confidence interval, HR = hazard ratio, MACE = major adverse cardiovascular event.

3.91; 3.68, 95% CI = 2.75–4.88; and 4.91, 95% CI = 3.80–6.26 respectively) than in the unchanged group (2.42, 95% CI = 2.02–2.90; 3.28, 95% CI = 2.81–3.83; and 4.72, 95% CI = 4.13–5.36, respectively) with an adjusted hazard ratio (aHR) of 1.20 (95% CI = 0.83–1.73), 1.12 (95% CI = 0.80–1.56) and 1.04 (95% CI = 0.78–1.39), respectively. The incidences of major bleeding between the two groups were lower (2.12, 95% CI = 1.43–3.01 for the de-escalated group and 2.36, 95% CI = 1.95–2.82 for unchanged group) and a significant difference between the two groups was not observed (adjusted HR of 0.92 [95% CI = 0.67–1.37]) (Table 2, Fig 3A). The incidence of non-major clinically relevant bleeding was increased in both groups; however, no significant differences were observed between the two groups (Fig 3B).

## Sensitivity analysis

To increase the validity of the study, we also performed sensitivity analyses that defined the occurrence of an event of interest based on primary diagnosis only. The results were consistent with the main findings (See S2 Table).

## Discussion

This current study demonstrated that approximately one in five patients adopted de-escalation of treatment to clopidogrel in the real-world data. These patients were older an elevated risk of bleeding even though they had a higher risk of cardiovascular events. We currently report no significant difference in cardiovascular and bleeding events within one year of follow-up after applying the IPTW technique to adjust baseline difference between the two experimental groups. These results may suggest that de-escalation of treatment to clopidogrel can be adopted in patients with a higher risk of bleeding.

Oral P2Y₁₂ receptor inhibitors are critical for secondary prevention of thrombotic events in ACS patients, particularly for those treated with PCI. Ticagrelor is superior to clopidogrel in preventing ischemic events due to its rapid onset and potent antiplatelet effects. The current clinical practice guidelines recommend the use of the novel P2Y₁₂ inhibitor, ticagrelor, as the first-line antiplatelet agent in ACS patients after PCI based on results from the PLATO trial in which ticagrelor-treated ACS patients had significantly lower rates of vascular death and MI than clopidogrel-treated patients [4].

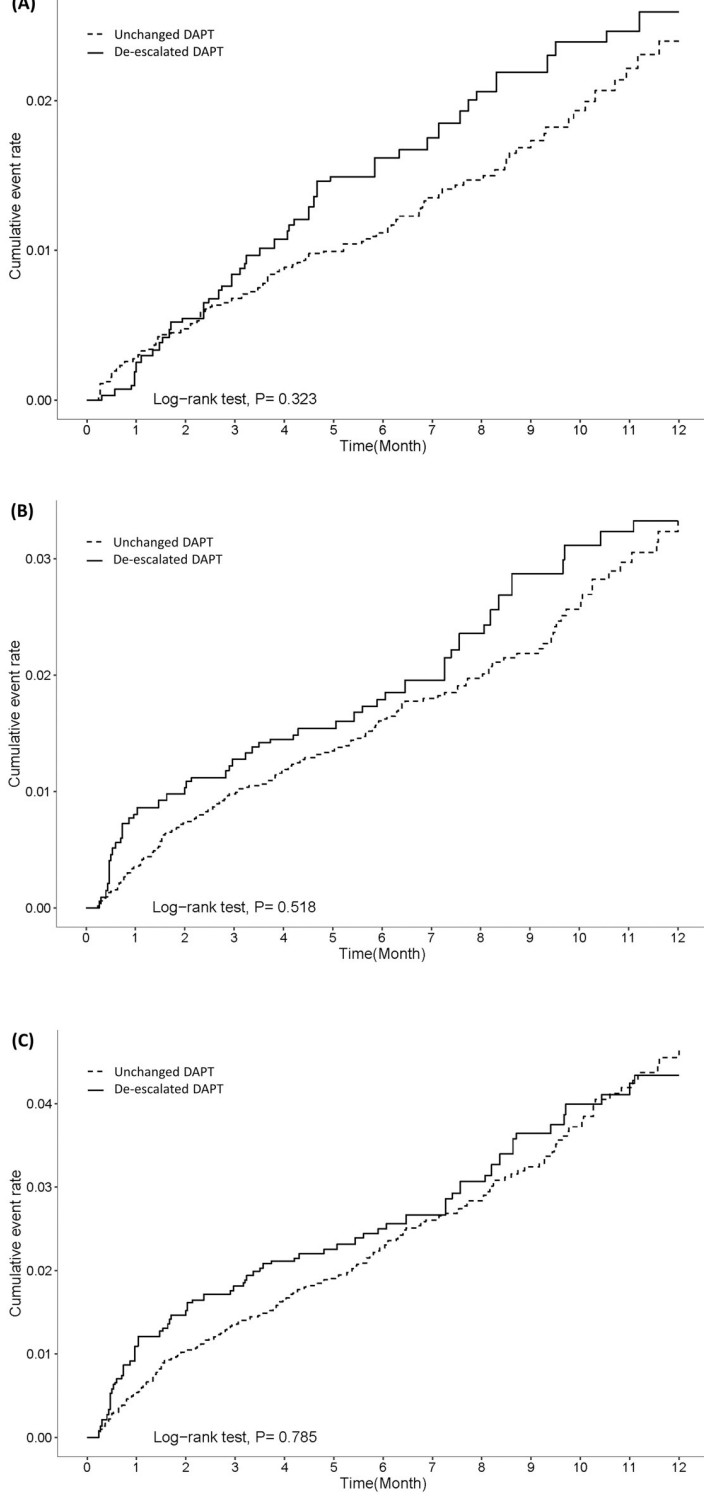

**Fig 2. Kaplan-Meier curves along with adjusted hazard risk for efficacy outcomes between unchanged and de-escalated groups.** A) all cause death. B) AMI readmission. C) MACE.

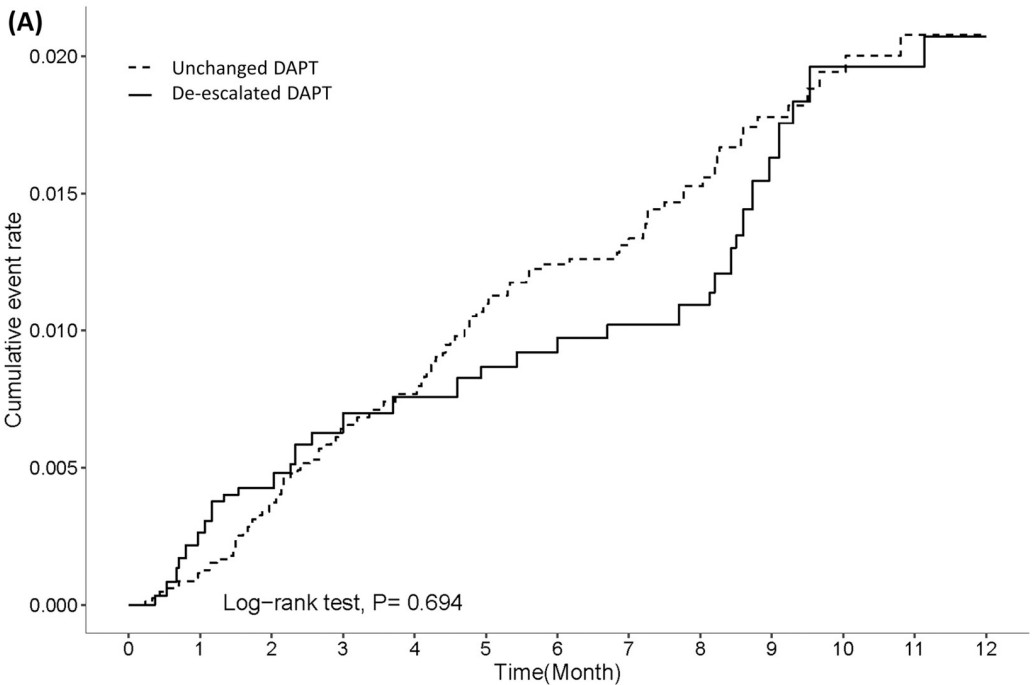

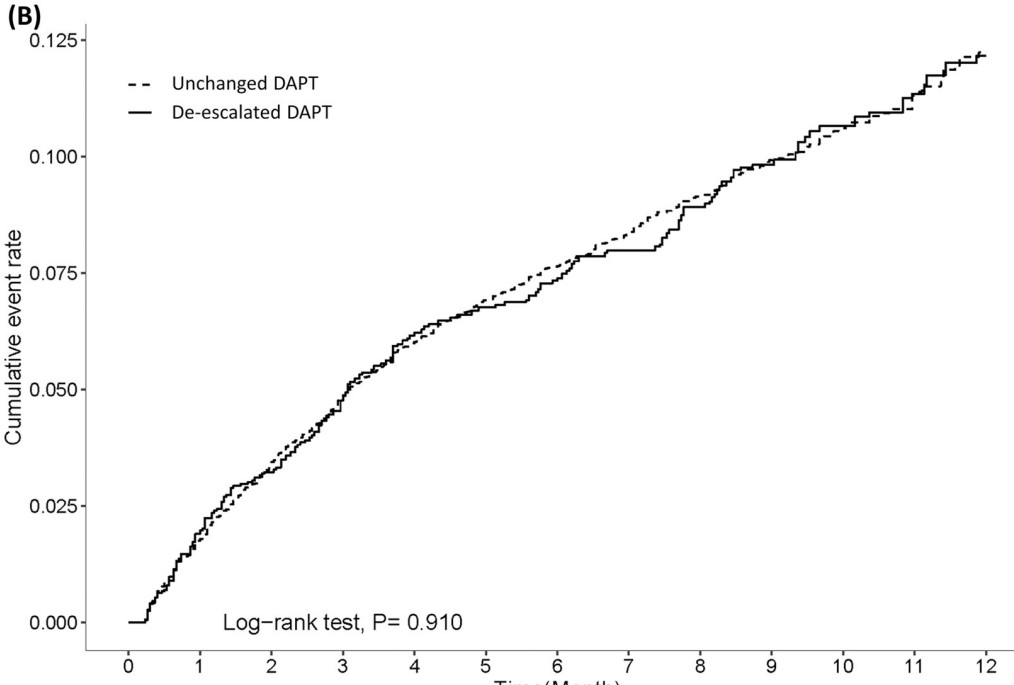

**Fig 3. Kaplan-Meier curves along with adjusted hazard risk for safety outcomes between unchanged and de-escalated groups.** A) Major bleeding. B) Non-major clinically relevant bleeding.

De-escalation from ticagrelor to clopidogrel occurs with a variable frequency in real-life practice that cannot be fully addressed in this current study. According to some clinical registries data, approximately 50% of ticagrelor cessation were the result of ticagrelor-specific adverse effects, and the remainder included various clinical scenarios, such as initiation of oral anticoagulation, unspecified preference of the treating physician, or financial reasons [14,15]. Regarding contemporary evidence of P2Y$_{12}$ de-escalation treatment, the results are inconsistent [16–19]. In the TOPIC study, ACS patients treated with PCI and after 1 month of DAPT with more potent P2Y$_{12}$ inhibitors were randomised to continue the treatment or switch to clopidogrel for 12 months. Compared with the de-escalated group, patients who continued treatment with potent P2Y$_{12}$ inhibitors had similar ischemic outcomes, but a significantly higher bleeding risk [6]. A pharmacodynamic study (SWAP-4) showed that de-escalation from ticagrelor to clopidogrel therapy was associated with an increase in platelet activity suggestive of a drug-drug interaction in the patient with stable coronary artery disease [20].

Meanwhile, the SCOPE registry (Switching From Clopidogrel to New Oral Antiplatelet Agents During Percutaneous Coronary Intervention), a prospective single center observational study with 1,363 ACS patients, suggested that de-escalation of antiplatelets early within 1 month after the index event in patients with acute coronary syndrome was associated with an increased risk of ischemic events (OR = 5.3; 95% CI: 2.1–18.2; p = 0.04) with no differences in bleeding at 1-month follow up [21]. In our study, the baseline characteristics of the de-escalation group revealed patients were older and with more comorbidities, which is aligned with the real-world practice. The IPTW method was also used to correct for possible selection bias. Although the other clinical circumstance and reasons for de-escalation P2Y12 inhibitors that were unable to be captured in the claim-based study. This population-based real-world data found that de-escalation from ticagrelor to clopidogrel 3 months after the index event of acute myocardial infarction had a comparable 1-year cardiovascular outcome with standard DAPT with ticagrelor treatment. The differences can be explained due to the heterogeneity of data source, study population, study design and so on.

The time point of de-escalation is also crucial since the ischemic risk is highest during the first 30 days of index event. A retrospective study that included 1,019 ACS patients with de-escalation P2Y$_{12}$ inhibitor treatment strategy showed that early de-escalation from ticagrelor to clopidogrel during the initial 30 days after ACS had an increased risk of ischemic events compared with switching beyond 30 days [22]. In this claim-based study, a 3-month window, which covered at least 2 outpatient visits, was used to identify whether their DAPT treatment scheme was switched from ticagrelor plus aspirin to clopidogrel plus aspirin. Based on our data (see KM curve Fig 2A), we did not observe the risk of AMI admission within 30 days in the de-escalation group. This provided some insights in the real-world clinical practice of the de-escalation DAPT strategy in stabilized AMI patients after PCI. Since the time point of de-escalation is very important for DAPT de-escalating strategy, we need more large-scale studies to investigate.

Unlike the TOPIC study results that reported the de-escalation strategy may be associated with a reduction of bleeding, the major bleeding and non-major clinically relevant bleeding rate were 2.1% and 14.3% in the de-escalation group, which was similar to the unchanged group. One possibility was the de-escalation group had a higher likelihood of bleeding in this retrospective cohort and the benefit of bleeding risk reduction from de-escalation strategy might have been neutralized. Another possibility was the bleeding risk may be stabilized 3 months after the AMI event in both groups. A large scale randomized controlled study should be launched to clarify the net-clinical benefit of a P2Y$_{12}$ inhibitor de-escalation strategy in ACS population.

In the ticagrelor-specific adverse effect, dyspnea is a second well-known side effect after bleeding. Nevertheless, rates of cessation because of dyspnea amounted to 5% in stable patients

treated in phase III off-label trails [23,24] and around 1% among patients with ACS in the randomized controlled trial and clinical registry [15], possibly because minor symptoms are better tolerated or may be attributed to other causes like cardiac or pulmonary cause than drug related effects in the early phase of ACS [25]. Unfortunately, this claim-based study was unable to address the association between ticagrelor and dyspnea.

Considering personalized medicine is getting more and more attention, some studies have been focused on the guided P2Y$_{12}$ inhibitor de-escalation strategy. The most popular is platelet function testing-guided or genetic-guided. In an open-label multicenter trial (TROPICA-L-ACS), Sibbling D et al. demonstrated platelet function testing guided de-escalation antiplatelet treatment is non-inferior to standard prasugrel treatment at 1 year after PCI in terms of net clinical benefit [26]. However, compared with an unguided de-escalation strategy, the cost-effectiveness of these guided strategies requires additional evaluation [27].

## Limitations

First, the NHIRD does not include all patient information, such as risk behaviours, diet and physical activities, which might be associated with the incidence of death or AMI readmission. Although this study used propensity score technique to balance the baseline difference between two groups, several important unmeasured confounding factors, as such frailty of patients, were unable to control due to the inherent limitation of administrative database. Thus, a hidden bias of medication selection was introduced. Second, the NHIRD does not contain clinical information, such as angiographic findings during PCI, the extent of coronary artery disease, and the severity of AMI at admission. Consequently, we could not adjust the severity of AMI nor could we identify whether the AMI admission was planned or not, which might induce non-differential misclassification bias. Finally, since we only included Taiwanese patients, the results might not be generalizable to other populations.

## Conclusions

Using Taiwan National Health Insurance Research Database, we evaluated the effect of de-escalation and unchanged of P2Y$_{12}$ inhibitor in dual antiplatelet therapy on major adverse cardiovascular events and bleeding complications for 1 year after AMI in patients undergoing successful percutaneous coronary intervention. There was no significant difference in the risk of death, AMI readmission, or MACE. Additionally, there was no difference in the risk of bleeding. A large-scale investigation is warranted to identify the profiles of patients suitable for de-escalation, the impact of de-escalation on adverse clinical outcomes, and to further compare a guided versus unguided de-escalation P2Y$_{12}$ inhibitor strategy.

## Supporting information

**S1 Fig.**
(TIF)

**S1 Table. Disease diagnosis codes according to ICD-9-CM and ATC classification of medications.**
(DOCX)

**S2 Table. The incidence (per 100 PY) and adjusted HR of major vascular and bleeding events in unchanged DAPT versus de-escalation DAPT group during one-year follow-up.**
(DOCX)

## Author Contributions

**Conceptualization:** Chun-Yao Huang, Yi-Chen Hsieh.

**Data curation:** Chun-Yao Huang, Wan-Ting Chen, Li-Nien Chien.

**Formal analysis:** Wan-Ting Chen, Yi-Chen Hsieh, Li-Nien Chien.

**Investigation:** Jong-Shiuan Yeh.

**Methodology:** Li-Nien Chien.

**Project administration:** Jong-Shiuan Yeh.

**Software:** Wan-Ting Chen.

**Supervision:** Chun-Yao Huang, Li-Nien Chien.

**Writing – original draft:** Jong-Shiuan Yeh.

**Writing – review & editing:** Jong-Shiuan Yeh, Chien-Yi Hsu, Li-Nien Chien.

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
