## [Decision Letter · Decision Letter 0]

27 Aug 2020

PONE-D-20-17922

The effect of de-escalation of P2Y12 receptor inhibitor therapy after acute myocardial infarction in patients undergoing percutaneous coronary intervention: a nationwide cohort study

PLOS ONE

Dear Dr. Chien,

Thank you for submitting your manuscript to PLOS ONE. After careful consideration, we feel that it has merit but does not fully meet PLOS ONE’s publication criteria as it currently stands. Therefore, we invite you to submit a revised version of the manuscript that addresses the points raised during the review process.

We look forward to receiving your revised manuscript.

Kind regards,

George C.M. Siontis, MD, PhD

Academic Editor

PLOS ONE

Journal Requirements:

2. Thank you for submitting the above manuscript to PLOS ONE. During our internal evaluation of the manuscript, we found significant text overlap between your submission and the following previously published works (text overlap mainly occurs in the abstract), some of which you are an author.

https://www.emjreviews.com/cardiology/abstract/the-effect-of-de-escalated-switching-dual-antiplatelet-therapy-after-acute-myocardial-infarction-in-patients-undergoing-percutaneous-coronary-intervention-real-world-data-from-a-nationwide-cohort-stu/

Please revise the manuscript to rephrase the duplicated text, cite your sources, and provide details as to how the current manuscript advances on previous work.

Please note that further consideration is dependent on the submission of a manuscript that addresses these concerns about the overlap in text with published work.

3. In the ethics statement in the manuscript and in the online submission form, please provide additional information about the patient records used in your retrospective study.

Specifically, please ensure that you have discussed whether all data were fully anonymized before you accessed them and/or whether the IRB or ethics committee waived the requirement for informed consent.

If patients provided informed written consent to have data from their medical records used in research, please include this information

Reviewers' comments:

Reviewer's Responses to Questions

**Comments to the Author**

1. Is the manuscript technically sound, and do the data support the conclusions?

Reviewer #1: Partly

Reviewer #2: Yes

2. Has the statistical analysis been performed appropriately and rigorously? 

Reviewer #1: Yes

Reviewer #2: Yes

3. Have the authors made all data underlying the findings in their manuscript fully available?

Reviewer #1: No

Reviewer #2: No

4. Is the manuscript presented in an intelligible fashion and written in standard English?

Reviewer #1: Yes

Reviewer #2: Yes

5. Review Comments to the Author

Reviewer #1: Yeh et al. report the impact of the de-escalation of P2Y12 inhibitors (i.e. ticagrelor to clopidogrel) on clinical outcomes among patients with STEMI. Among 10,100 STEMI patients (1901 with de-escalation and 8199 without de-escalation) from the Taiwan’s National Health Insurance Research Database, after IPTW adjustment, there were no significant differences in death (HR 1.20, 95% CI 0.83-1.73), AMI readmission (HR 1.12, 95% CI 0.80-1.56), MACE (HR 1.04, 95% CI 0.78–1.39), and major bleeding (HR 0.92, 95% CI 0.67-1.37) between groups.

This reviewer has the following comments:

- The clinical circumstance and reason for switching, relevant factors for patients’ outcome, were not reported in the current study. Considering that some of the clinical events (e.g. bleeding) triggered switching, only events occurred after switching should be considered for analysis (at least as a sensitivity analysis) to assess the causal relationship between the de-escalation and subsequent clinical events.

- No significant decrease in bleeding events in the de-escalation group may be at least partly due to the patient characteristics not captured in the administrative database (e.g. frailty, risk profile change after index AMI). Several important unmeasured confounding factors due to the inherent limitation of administrative database may hinder to draw a definitive conclusion.

Methods

- Why did the authors exclude patients treated with DAPT with prasugrel?

- Is there validation data on AMI readmission, CV death, and bleeding in the Taiwan’s National Health Insurance Research Database? If not available, please comment on the accuracy of data.

- Please explain why the 3-month window was chosen.

- Please provide the rational for the exclusion of CABG patients.

- IPTW adjustment: Was propensity score used for the IPTW adjustment? Medication possession ratio may include data after switching from ticagrelor to clopidogrel, which should not be used for the calculation of propensity score. Please provide the distribution of propensity score between 2 groups.

- Definition of AMI readmission “the patient receiving a primary or secondary diagnosis of AMI after the index of AMI admission”: Including secondary diagnosis of AMI may limit the accuracy of AMI readmission compared with including only a primary diagnosis of AMI readmission.

- Please provide the definition of CV death.

- Please explain the rationale for using SMD to assess differences between groups instead of P values.

Results

- It seems to be not necessary to show baseline characteristics after IPTW adjustment.

- Please provide the mean follow-up duration.

- Please provide the clinical outcomes as a table.

Reviewer #2: Jong Shiuan Yeh et al. addressed the highly interesting field of de-escalating potent P2Y12 inhibitors in acute myocardial infarction patients post-PCI. The presented data is of high quality and contains more than 10’000 patients of which 1’901 switched to clopidogrel and 8’199 remained on ticagrelor. The Authors found no difference in ischemic events (all cause death, AMI readmission, and MACE) within one year follow up. Similarly, the incidences of major bleeding between the two groups was not different. The authors concluded correctly that a more detailed investigation is needed to answer the present question, yet the contemporary trial suggest no harm nor benefit. The Authors should be congratulated for their nice and accurate work.

The Present Reviewer has following questions

Major

• The timepoint of de-escalation is crucial since the ischemic risk is highest during the first 30 days. How was this implemented in the present analysis

• Different Reasons for de-escalation inflict different clinical scenarios and hitherto different risk. Was an investigation into reasons for de-escalation performed?

• There is an ongoing discussion about ticagrelor and dyspnea and its impact on real world adherence. Would the authors be able to address this question?

• Why were patients who died within the first 3 months after hospital discharge excluded?

• The SCOPE registry should be discussed in view of the present findings since it found higher events after de-escalation.

• The SWAP-4 trial, which assesses drug interaction due to de-escalation from ticagrelor to clopidogrel should be discussed.

Minor

• Number of patients at risk in the figures should be included

6. PLOS authors have the option to publish the peer review history of their article (what does this mean?). If published, this will include your full peer review and any attached files.

Reviewer #1: No

Reviewer #2: No

---

## [Author Response · Author response to Decision Letter 0]

3 Nov 2020

Reviewer #1: Yeh et al. report the impact of the de-escalation of P2Y12 inhibitors (i.e. ticagrelor to clopidogrel) on clinical outcomes among patients with STEMI. Among 10,100 STEMI patients (1901 with de-escalation and 8199 without de-escalation) from the Taiwan’s National Health Insurance Research Database, after IPTW adjustment, there were no significant differences in death (HR 1.20, 95% CI 0.83-1.73), AMI readmission (HR 1.12, 95% CI 0.80-1.56), MACE (HR 1.04, 95% CI 0.78–1.39), and major bleeding (HR 0.92, 95% CI 0.67-1.37) between groups. This reviewer has the following comments:

1. The clinical circumstance and reason for switching, relevant factors for patients’ outcome, were not reported in the current study. Considering that some of the clinical events (e.g. bleeding) triggered switching, only events occurred after switching should be considered for analysis (at least as a sensitivity analysis) to assess the causal relationship between the de-escalation and subsequent clinical events.

Response: Thank you for your comment and suggestion. This current study was aimed to study the subsequent outcomes after de-escalating of P2Y12 inhibitors among STEMI patients. We understood that several clinical circumstance and reasons for switching were unable to capture in the claim-based study although we used the propensity score technique to reduce the potential bias. We addressed this limitation in the revised manuscript. We also provided a sensitivity analysis that included the patients occurred an event of bleeding before switching. We can only identify 117 patients (<10% of switch group) had a diagnostic code of bleeding before de-escalation and their bleeding risk was significantly higher than those without de-escalation of P2Y12 inhibitors while no different in the risk of AMI admission and MACE. 

Page 29, line 264 Considering that some of the clinical events (e.g. bleeding) triggered switching, we conducted a sensitivity analysis that only included the patients occurred an event of bleeding before switching. We can only identify 117 patients (<10% of de-escalation group) had a diagnostic code of bleeding before de-escalation and their bleeding risk was significantly higher than those without de-escalation of P2Y12 inhibitors while no different in the risk of AMI admission and MACE (See appendix table 3). 

2. No significant decrease in bleeding events in the de-escalation group may be at least partly due to the patient characteristics not captured in the administrative database (e.g. frailty, risk profile change after index AMI). Several important unmeasured confounding factors due to the inherent limitation of administrative database may hinder to draw a definitive conclusion.

Response: Thanks for the comments. We recognized that some factors associated with de-escalation of P2Y12 inhibitors and outcomes were unmeasured due to the inherent limitation of administrative database. These factors might be biased our finding. We mentioned the limitation in the discussion section. 

Page 33, line 319. Limitations. Although this study used propensity score technique to balance the baseline difference between two groups, several important unmeasured confounding factors, as such frailty of patients, were unable to control due to the inherent limitation of administrative database. Thus, a hidden bias of medication selection was introduced. 

Methods

3. Why did the authors exclude patients treated with DAPT with prasugrel?

Response: We did not include the patients treated with DAPT with prasugrel because this medication was not approval for reimbursement under the regulation of National Health Insurance in Taiwan during the study period (here is from Jan 2013 to Dec 2016). Thus, we did not consider to include the patients treated with DAPT with prasugrel. 

Page 6, line 105. The patients treated with prasugrel were not included because it had not been approved for reimbursement under the regulation of National Health Insurance during the study period. 

4. Is there validation data on AMI readmission, CV death, and bleeding in the Taiwan’s National Health Insurance Research Database? If not available, please comment on the accuracy of data.

Response: Thanks for your suggestion. The accuracy of diagnostic code of AMI admission has been validated with a high PPV of 0.88 in any diagnosis based on NHIRD (Cheng et al., 2014). However, we did not find the validation data of bleeding. Fortunately, a validation study conducted in Korea found that the PPV of primary diagnostic code of GI bleeding in Health Insurance Claim was over 90%, showing favorable reliability (Park et al., 2019). The data of CV death was derived from National Death Registry (NDR). The completeness and accuracy of death records in Taiwan were high(Lu, Lee, & Chou, 2000), as it is mandatory to register all causes of death in the NDR. We added the information regarding the accuracy of data in the method section. 

Page 8, line 148. The accuracy of diagnostic code of AMI admission has been validated with a high PPV of 0.88 in any diagnosis based on NHIRD (Cheng et al., 2014). In this current study, we additionally combined antiplatelet therapy to confirm the AMI diagnosis was active. The accuracy of bleeding diagnosis has not been validated in NHIRD; however, a validation study conducted in Korea found that the PPV of primary diagnostic code of GI bleeding in Health Insurance Claim was over 90%, showing favorable reliability(Park et al., 2019). The data of CV death was derived from National Death Registry (NDR). The completeness and accuracy of death records in Taiwan were high (Lu et al., 2000), as it is mandatory to register all causes of death in the NDR.

5. Please explain why the 3-month window was chosen.

Response: Thanks for your suggestion. The timing of de-escalating of P2Y12 inhibitors mainly depends on the treatment scheme after AMI discharge. A 3-month window was chosen because we can observe at least two outpatient visits in the period, allowing de-escalating of P2Y12 inhibitors if necessary. 

Page 6, line 108. In the analytic cohort, a 3-month window was used to identify whether their DAPT treatment scheme was switched from ticagrelor plus aspirin to clopidogrel plus aspirin. A 3-month window was chosen because we can observe at least two outpatient visits in the period, allowing de-escalating of P2Y12 inhibitors if necessary. 

6. Please provide the rational for the exclusion of CABG patients.

Response: The exclusion of CABG patients was because these patients usually were not treated DAPT with either ticagrelor or clopidogrel in our study period (2013-2016). As we know, the ACC/AHA guideline (Levine et al., 2016) focused update provided a class I recommendation for the resumption of DAPT after CABG in patients with acute coronary syndrome based on level C evidence in 2016. Despite this, the DAPT prescription in discharged patients with ACS post-CABG is still low and varied. Thus, we excluded those patients to increase the homogeneity in study sample. The rational for the exclusion of CABG was provided in the revised manuscript. 

Page 6, line 99. 3) patients had undergone a coronary artery bypass graft during the study period because the DAPT prescription with either ticagrelor or clopidogrel in these patients were low despite the guidelines recommended (Levine et al., 2016). We therefore excluded these patients to increase the homogeneity of the study sample.

7. IPTW adjustment: Was propensity score used for the IPTW adjustment? Medication possession ratio may include data after switching from ticagrelor to clopidogrel, which should not be used for the calculation of propensity score. Please provide the distribution of propensity score between 2 groups.

Response: Thanks for your comment and suggestion. The distribution of propensity score between 2 groups is provided in the revised manuscript in the supplementary material. The propensity score was used for the IPTW adjustment in this current study. We considered the medication possession ratio (MPR) within three months into the propensity score because the efficacy of medication can be associated the subsequent health outcomes. 

8. Definition of AMI readmission “the patient receiving a primary or secondary diagnosis of AMI after the index of AMI admission”: Including secondary diagnosis of AMI may limit the accuracy of AMI readmission compared with including only a primary diagnosis of AMI readmission.

Response: Thanks for your suggestion. In the current study, we used any diagnosis of AMI admission because the accuracy of diagnosis was high (Cheng et al., 2014). As suggested, we conducted a sensitivity analysis that defined the occurrence of an event of interest based on the primary diagnosis only. The results were consistent with our main finding and we added the information in the revised manuscript. 

Page 26, line 221. To increase the validity of the study, we also performed sensitivity analyses that defined the occurrence of an event of interest based on primary diagnosis only. The results were consistent with the main findings (See supplementary Table 2). 

9. Please provide the definition of CV death.

Response: The definition of CV death has been included in the Supplementary Table 1. 

10. Please explain the rationale for using SMD to assess differences between groups instead of P values.

Response: Thanks for your suggestion. Standardized mean difference (SMD) is the most commonly used statistic to examine the balance of covariate distribution between treatment groups in the propensity score analysis. Because SMD is independent of the unit of measurement, it allows comparison between variables with different unit of measurement(Austin, 2010). Thus, we provided the rationale in the revised manuscript.

Page 18, line 158. Baseline characteristics were analyzed using standardized mean differences (SMD). SMD is the most commonly used statistic to examine the balance of covariate distribution between treatment groups in the propensity score analysis. Because SMD is independent of the unit of measurement, it allows comparison between variables with different unit of measurement. An SMD of >0.1 indicated the presence of a non-negligible difference between the two groups(Austin, 2010).

Results

11. It seems to be not necessary to show baseline characteristics after IPTW adjustment.

Response: Thanks for your comment. Since we would like to show the baseline characteristic between two groups can be balanced after the IPTW. Thus, we would like to present the baseline difference before and after IPTW.

12. Please provide the mean follow-up duration.

Response: The data of the mean follow-up duration has been reported in Table 1 in the revised manuscript. 

13. Please provide the clinical outcomes as a table.

Response: The data of the clinical outcomes has been reported in Table 2 in the revised manuscript (see Page 24-Page 25).

Reviewer #2: Jong Shiuan Yeh et al. addressed the highly interesting field of de-escalating potent P2Y12 inhibitors in acute myocardial infarction patients post-PCI. The presented data is of high quality and contains more than 10’000 patients of which 1’901 switched to clopidogrel and 8’199 remained on ticagrelor. The Authors found no difference in ischemic events (all cause death, AMI readmission, and MACE) within one year follow up. Similarly, the incidences of major bleeding between the two groups was not different. The authors concluded correctly that a more detailed investigation is needed to answer the present question, yet the contemporary trial suggest no harm nor benefit. The Authors should be congratulated for their nice and accurate work. The Present Reviewer has following questions. 

Major

1. The time point of de-escalation is crucial since the ischemic risk is highest during the first 30 days. How was this implemented in the present analysis

Response: Thank you for your comment. We understand that the time point of de-escalation is important in this study. Unlike the RCT, it is difficult to define the exact time of DAPT de-escalation in the claim-based research. In our study, a 3-month window, which covered at least 2 outpatient visits, was used to identify whether their DAPT treatment scheme was switched from ticagrelor plus aspirin to clopidogrel plus aspirin. We have made some description in the discussion section.

Page 30, line 282. The time point of de-escalation is crucial since the ischemic risk is highest during the first 30 days of index event. A retrospective study that included 1,019 ACS patients with de-escalation P2Y12 inhibitor treatment strategy showed that early de-escalation from ticagrelor to clopidogrel during the initial 30 days after ACS had an increased risk of ischemic events compared with switching beyond 30 days. [21]In this claim-based study, a 3-month window, which covered at least 2 outpatient visits, was used to identify whether their DAPT treatment scheme was switched from ticagrelor plus aspirin to clopidogrel plus aspirin. Based on our data (see KM curve figure 2A), we did not observe the risk of AMI admission within 30 days in the de-escalation group. Since the time point of de-escalation is very important for patients de-escalating potent P2Y12 inhibitors, future study is worth to investigate. 

2. Different Reasons for de-escalation inflict different clinical scenarios and hitherto different risk. Was an investigation into reasons for de-escalation performed?

Response: Thanks for your comments. There are several clinical circumstance and reasons for de-escalation P2Y12 inhibitors that were unable to capture in the claim-based study. We acknowledged it in the discussion section. 

Page 28 line 244. De-escalation from ticagrelor to clopidogrel occurs with a variable frequency in real-life practice that cannot be fully addressed in this current study According to some clinical registries data, approximately 50% of ticagrelor cessation were the result of ticagrelor-specific adverse effects, and the remainder included various clinical scenarios, such as initiation of oral anticoagulation, unspecified preference of the treating physician, or financial reasons. {Fosbøl, 2016 #113} {Zanchin, 2018 #104} 

3. There is an ongoing discussion about ticagrelor and dyspnea and its impact on real world adherence. Would the authors be able to address this question?

Response: In this current study, we were unable to address the association between ticagrelor and dyspnea since the side effect of medication is usually underestimated in claim database. Besides, it is very difficult to distinguish the risk of dyspnea was due to medication use or disease condition itself. We would like to address the issue in the discussion section in the revised manuscript.

Page 32, line 301. In the ticagrelor-specific adverse effect, dyspnea is a second well-known side effect after bleeding. Nevertheless, rates of cessation because of dyspnea amounted to 5% in stable patients treated in phase III off-label trails {Hiatt, 2017 #108} {Bonaca, 2015 #107} and around 1% among patients with ACS in the randomized controlled trial and clinical registry {Zanchin, 2018 #104} , possibly because minor symptoms are better tolerated or may be attributed to other causes like cardiac or pulmonary cause than drug related effects in the early phase of an ACS{Granger, 2016 #109}. Unfortunately, this current study was unable to address the association between ticagrelor and dyspnea. 

4. Why were patients who died within the first 3 months after hospital discharge excluded?

Response: We excluded the patients who died within the first 3 months after hospital discharge since their disease conditions and comorbidities were more likely to be complicated, resulting in the treatment scheme unlikely following the recommended guidelines. Besides, we used a 3-month window to group patients into de-escalation cohort, resulting in being unable to classify these patients. Thus, we excluded this population. 

Page 8 line 102. The later exclusion was made because their disease condition and comorbidities were more likely to be complicated; thus, their treatment were less likely to follow the recommended guidelines. Besides, we used a 3-month window to group patients into de-escalation cohort, resulting in being unable to classify these patients. Thus, we excluded this population.

5. The SCOPE registry should be discussed in view of the present findings since it found higher events after de-escalation.

Response: Thanks for your comment. We added the discussion in the revised manuscript. 

Page 29, line 257. Meanwhile, the SCOPE registry (Switching From Clopidogrel to New Oral Antiplatelet Agents During Percutaneous Coronary Intervention), a prospective single center observational study with 1363 ACS patients, suggested that de-escalation of antiplatelets early within 1 month after the index event in patients with acute coronary syndrome was associated with an increased risk of ischemic events (OR = 5.3; 95% CI: 2.1-18.2; p = 0.04) with no differences in bleeding at 1-month follow up. However, this population-based real-world data found that de-escalation from ticagrelor to clopidogrel 3 months after the index event of acute coronary syndrome was not associate with poor cardiovascular outcomes. The differences can be explained due to the heterogeneity of data source, study population, study design and so on. 

6. The SWAP-4 trial, which assesses drug interaction due to de-escalation from ticagrelor to clopidogrel should be discussed.

Response: Thank for your comment. The SWAP-4 trial has been summarized and discussed in the revised manuscript. 

Page 29, line 254. A pharmacodynamic study (SWAP-4) showed that de-escalation from ticagrelor to clopidogrel therapy was associated with an increase in platelet activity suggestive of a drug-drug interaction in the patient with stable coronary artery disease.

Minor 

7. Number of patients at risk in the figures should be included

Response: This study used IPTW approach to adjust the baseline difference between two groups. The propensity score is used as a weight index; as a result, the number of sample size in each group was not an integer and the sample size were not the same as in two study cohorts before matching. Thus, we decided not to report the numbers at risk to avoid the confusing. 

Austin, P. C. (2010). The performance of different propensity-score methods for estimating differences in proportions (risk differences or absolute risk reductions) in observational studies. Stat Med, 29(20), 2137-2148. doi:10.1002/sim.3854

Cheng, C.-L., Lee, C.-H., Chen, P.-S., Li, Y.-H., Lin, S.-J., & Yang, Y.-H. K. (2014). Validation of acute myocardial infarction cases in the national health insurance research database in taiwan. Journal of Epidemiology, 24(6), 500-507. doi:10.2188/jea.je20140076

Levine, G. N., Bates, E. R., Bittl, J. A., Brindis, R. G., Fihn, S. D., Fleisher, L. A., . . . Wijeysundera, D. N. (2016). 2016 ACC/AHA guideline focused update on duration of dual antiplatelet therapy in patients with coronary artery disease: A report of the American College of Cardiology/American Heart Association Task Force on Clinical Practice Guidelines. J Thorac Cardiovasc Surg, 152(5), 1243-1275. doi:10.1016/j.jtcvs.2016.07.044

Lu, T. H., Lee, M. C., & Chou, M. C. (2000). Accuracy of cause-of-death coding in Taiwan: types of miscoding and effects on mortality statistics. Int J Epidemiol, 29(2), 336-343. doi:10.1093/ije/29.2.336

Park, J., Kwon, S., Choi, E.-K., Choi, Y.-j., Lee, E., Choe, W., . . . Oh, S. (2019). Validation of diagnostic codes of major clinical outcomes in a National Health Insurance database. International Journal of Arrhythmia, 20(1), 5. doi:10.1186/s42444-019-0005-0

---

## [Decision Letter · Decision Letter 1]

10 Dec 2020

PONE-D-20-17922R1

The effect of de-escalation of P2Y12 receptor inhibitor therapy after acute myocardial infarction in patients undergoing percutaneous coronary intervention: a nationwide cohort study

PLOS ONE

Dear Dr. Chien,

Thank you for submitting your manuscript to PLOS ONE. After careful consideration, we feel that it has merit but does not fully meet PLOS ONE’s publication criteria as it currently stands. Therefore, we invite you to submit a revised version of the manuscript that addresses the points raised during the review process.

We look forward to receiving your revised manuscript.

Kind regards,

George C.M. Siontis, MD, PhD

Academic Editor

PLOS ONE

Additional Editor Comments (if provided):

A couple of remaining issues previously raised by the reviewers need to be addressed.

Reviewers' comments:

Reviewer's Responses to Questions

**Comments to the Author**

1. If the authors have adequately addressed your comments raised in a previous round of review and you feel that this manuscript is now acceptable for publication, you may indicate that here to bypass the “Comments to the Author” section, enter your conflict of interest statement in the “Confidential to Editor” section, and submit your "Accept" recommendation.

Reviewer #1: (No Response)

2. Is the manuscript technically sound, and do the data support the conclusions?

Reviewer #1: Yes

3. Has the statistical analysis been performed appropriately and rigorously? 

Reviewer #1: Yes

4. Have the authors made all data underlying the findings in their manuscript fully available?

Reviewer #1: No

5. Is the manuscript presented in an intelligible fashion and written in standard English?

Reviewer #1: Yes

6. Review Comments to the Author

Reviewer #1: This reviewer has the following comments:

- In the first revision round, this reviewer asked to provide the results of IPTW analysis for endpoints occurred after de-escalation of DAPT (i.e. patients with any events occurred before de-escalation should be censored) to assess the impact of the de-escalation on the subsequent clinical events. However, the authors provided the results in patients with bleeding before de-escalation. And it is obvious that those patients had a higher bleeding risk than those without de-escalation. If time points of events and DAPT de-escalation are available in the dataset, please provide aforementioned analysis instead of supplementary table 3.

- For better understanding of the readers, the definition of grouping should be clearly mentioned in the Method section (e.g. patients were considered the de-escalation group if ticagrelor was changed to clopidogrel 3 months after index AMI hospitalization).

7. PLOS authors have the option to publish the peer review history of their article (what does this mean?). If published, this will include your full peer review and any attached files.

Reviewer #1: No

---

## [Author Response · Author response to Decision Letter 1]

5 Jan 2021

n the first revision round, this reviewer asked to provide the results of IPTW analysis for endpoints occurred after de-escalation of DAPT (i.e. patients with any events occurred before de-escalation should be censored) to assess the impact of the de-escalation on the subsequent clinical events. However, the authors provided the results in patients with bleeding before de-escalation. And it is obvious that those patients had a higher bleeding risk than those without de-escalation. If time points of events and DAPT de-escalation are available in the dataset, please provide aforementioned analysis instead of supplementary table 3.

Response: Thank for the comment. We are sorry for the misinterpretation of the reviewer’s suggestion in the last revision. In the study, the endpoints of any events were after de-escalation of DAPT. We have made more clarification in the method section and we had deleted the paragraph about supplementary table 3. 

Page 8 line 147-149. 

This study assessed the effect of DAPT de-escalation strategy on the subsequent endpoints, therefore the patients with any event occurred before the 3 months DAPT de-escalation time point was censored.

- For better understanding of the readers, the definition of grouping should be clearly mentioned in the Method section (e.g. patients were considered the de-escalation group if ticagrelor was changed to clopidogrel 3 months after index AMI hospitalization).

Response: Thank for the suggestion. We have made the definition clearer in the revision. 

Page 6 line 112-113.

The patients were considered as the de-escalation group if ticagrelor was changed to clopidogrel within 3 months after index AMI hospitalization.

---

## [Editor Report · Decision Letter 2]

13 Jan 2021

The effect of de-escalation of P2Y12 receptor inhibitor therapy after acute myocardial infarction in patients undergoing percutaneous coronary intervention: a nationwide cohort study

PONE-D-20-17922R2

Dear Dr. Chien,

We’re pleased to inform you that your manuscript has been judged scientifically suitable for publication and will be formally accepted for publication once it meets all outstanding technical requirements.

Kind regards,

George C.M. Siontis, MD, PhD

Academic Editor

PLOS ONE

---

## [Editor Report · Acceptance letter]

15 Jan 2021

PONE-D-20-17922R2 

The effect of de-escalation of P2Y_12_ receptor inhibitor therapy after acute myocardial infarction in patients undergoing percutaneous coronary intervention: a nationwide cohort study 

Dear Dr. Chien:

I'm pleased to inform you that your manuscript has been deemed suitable for publication in PLOS ONE. Congratulations! Your manuscript is now with our production department. 

Kind regards, 

on behalf of

Dr. George C.M. Siontis 

Academic Editor

PLOS ONE